# Rapid Measurements and Phase Transition Detections Made Simple by AC-GANs

Jiewei Ding[1,2], Ho-Kin Tang[3] * and Wing Chi Yu[1,2] †

**1** Department of Physics, City University of Hong Kong, Kowloon, Hong Kong
**2** City University of Hong Kong Shenzhen Research Institute, Shenzhen 518057, China
**3** School of Science, Harbin Institute of Technology, Shenzhen, 518055, China
* denghaojian@hit.edu.cn † wingcyu@cityu.edu.hk

January 25, 2024

## Abstract

In recent years, the use of end-to-end neural networks to analyze Monte Carlo data has received a lot of attention. However, the application of non-end-to-end generative adversarial neural networks is less explored. Here, we study classical many-body systems using generative adversarial neural networks. We use the conditional generative adversarial network with an auxiliary classifier (AC-GAN) and introduce self-attention layers into the generator, enabling the model to learn the distribution of two-dimensional XY model spin configurations as well as the physical quantities. By applying the symmetry of the systems, we further find that AC-GAN can be trained with a very small raw dataset, allowing us to obtain reliable measurements in the model that requires a large sample size, e.g. the large-sized 2D XY model and the 3D Heisenberg model. We also find that it is possible to quantify the distribution changes that occur in the configuration of the models during phase transitions and locate the phase transition points by AC-GAN.

## 1   Introduction

In condensed matter systems, a large number of microscopic particles interacting in the lattice can induce interesting collective physical phenomena [1]. Some of these phenomena give us hints to develop novel materials with useful properties. However, when we study many-body systems, we usually face difficulty in doing exact calculations due to the large coupled degree of freedoms. The stochastic nature of Monte Carlo (MC) simulation has made it an important tool for the study [2, 3]. Nevertheless, Monte Carlo methods possess problems such as critical slowdown around phase transitions that could hinder its applicability to complex systems. Improving the Monte Carlo algorithm or finding alternative tools to replace it has been an important active research area in the field [4–10].

As a statistical tool, deep learning has attracted lots of attention in physics and other branches of science in recent years because of its ability to fit arbitrary complex functions [11–17]. Using Monte Carlo simulation to study many-body systems can generate a large amount of high-quality data, therefore how to use deep learning to analyse these data has become a topic of interest. For example, previous work has shown that unsupervised deep learning can roughly locate the phase transition points in many-body systems through configuration samples generated by Monte Carlo simulation [18–20], while supervised learning can locate the transition points with high precision [21, 22]. There are also efforts in using deep learning to find the effective non-interacting system of a many-body system and introduce this subsystem in Monte Carlo simulation to speed up the sampling of the original system [8].

Generative Adversarial Networks (GAN) is a non-end-to-end deep learning model that has attracted attention for its ability to generate high-quality samples from an implicit probability distribution. In computer vision, GANs can be used to modify images, create images, and increase image resolution [23–25]. In recent years, GANs have also begun to gain attention in fields such as chemistry, pharmaceuticals, and engineering [26–31], while in physics, recent work by Japneet Singh found that implicit-GAN can replace Monte Carlo to sample the XY model spin configurations and predict the phase transition point without prior knowledge of symmetry breaking [32]. However, there exists differences between the spin configurations sampled by implicit-GAN and that sampled by Markov Chain Monte Carlo (MCMC), which cause the measured physical quantities to deviate in high-temperatures. In this work, we further explore the application of GAN to many-body systems and focus on solving the problems mentioned.

Firstly, to improve the performance of GAN in fitting Monte Carlo data, we use Conditional GAN with an auxiliary classifier (AC-GAN), which is easier to converge, as our main architecture [33]. To let the Generator performs better in learning the correlation between spin and spin, multi-head self-attention layers are inserted into the Generator. This results in the physical quantities measured by AC-GAN agree almost perfectly with those measured by MCMC [34, 35]. Secondly, we found that an AC-GAN can be successfully trained with an augmentation dataset obtained by applying rotation, translation, mirror symmetry, etc. on a

very small raw dataset. This allows us to apply AC-GAN to some many-body systems where sampling the spin configurations with MCMC is difficult. Finally, we find that GAN can be used to locate the phase transition point of many-body systems. The existing deep learning models used to classify phases of matter are still regarded as black boxes. They cannot explain what happens in the many-body system at the phase transition. In this work, we proposed a measure of the difference between the distribution of GAN-generated spin configuration samples and the Boltzmann distribution to locate the critical points based on the observation that the phase transition is accompanied by a significant change of configuration distribution. In this aspect, it is worth noting that our method does not require any pre-processing of the raw data, and is more interpretable than other methods that treat deep learning as a black box to locate phase transitions [18–22].

The paper is organized as follows. In Sec. 2, we introduce GAN and AC-GAN. We then compare the results in the two-dimensional (2D) XY model with configurations sampled by AC-GAN and MCMC in Sec.3. Sec. 4 shows the results of 2D XY model configurations and three-dimensional (3D) Heisenberg model configurations trained with data augmentation. Section 5 introduces the generative configurations property in GANs and shows how we can use this property to locate the phase transition of the 2D Ising model and the 2D XY model. Finally, a conclusion is given in Sec.6.

## 2   Conditional GAN with auxiliary classifier

The basic structure of GANs composes of a generator(G) and a discriminator(D), which are two independent deep learning models, and a database for storing the real samples. The input to the generator is a random matrix $z$ and the output are the generated samples. The generated samples together with the real samples are then input to the discriminator which outputs a binary classification that classifies the input into the generated and the real samples. The target of the discriminator is to best distinguish between the generated and real samples, while that for the generator is to produce samples as close as the real samples in order to fool the discriminator [23]. In practice, the discriminator and the generator are trained alternately and such an adversarial training allows the two models to improve their performances in parameter updates repeatedly.

In AC-GAN, besides the random matrix $z$, constrains are added in form of a conditional matrix $c$ in the input to the generator. The discriminator now has to simultaneously determine whether the input is a real sample or a generated sample, as well as which conditional class that the input sample belongs to. This extra output can better guide the discriminator on updating the machine's parameters during backward propagation which in turn results in a more stable training and faster convergence [33]. A schematic diagram of the AC-GAN architecture is shown in Fig.1.

For the many-body systems we considered in the following, we took the spin configurations at each site to form the input matrices and divided the continuous temperature range at an interval of 0.2 to form $N_c$ conditional classes. In general, the elements in the configuration matrix are correlated as spatial correlations are present in the many-body system [36–38]. However, the kernel in the convolutional neural network (CNN) in a classical generator has a small receptive field (the common kernel size is 3x3 in which the effective receptive field is only 2x2) [39]. This is disadvantageous for the generator to produce configurations with long correlation lengths. To improve the performance, we introduced the self-attention layers to

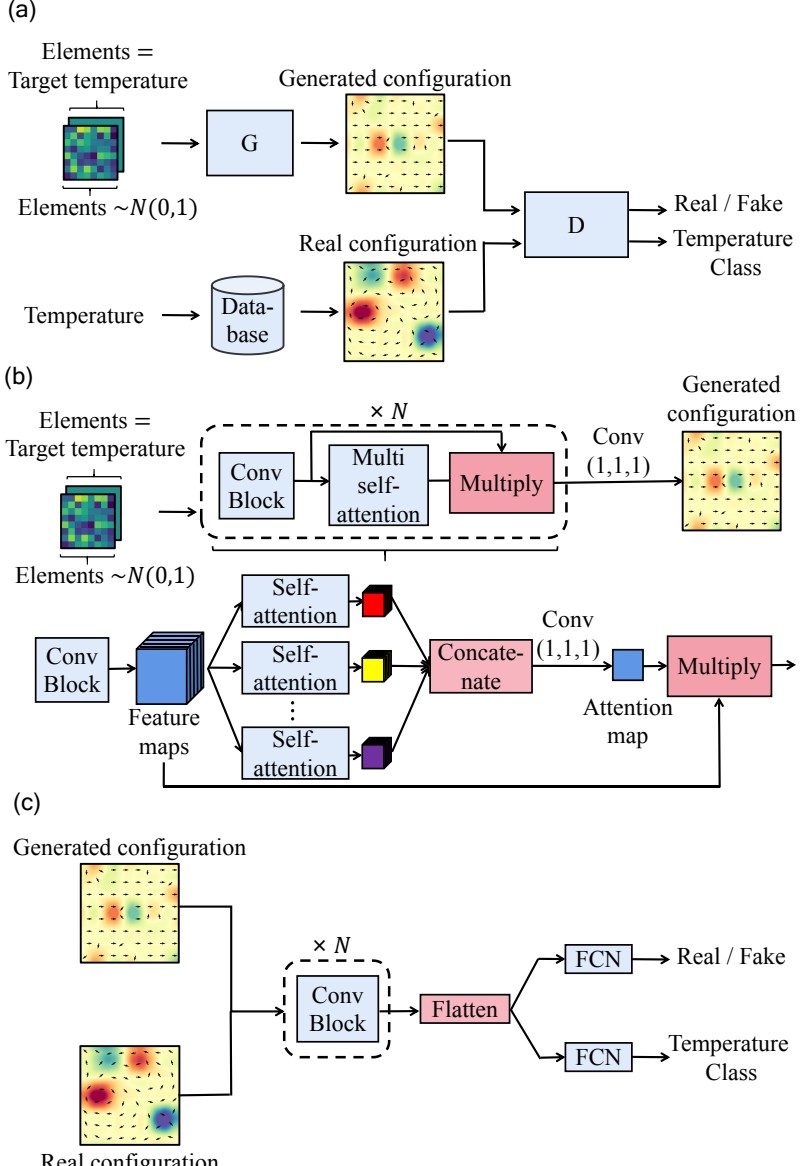

Figure 1: (a) shows the main architecture of the AC-GAN. The input to the generator are a noise matrix with elements generated from a Gaussian distribution and a condition matrix. (b) and (c) show the architecture of the generator (G) and the discriminator (D), respectively. The generator composes of a series of CNN blocks and self-attention blocks while the discriminator consists of a series of CNN blocks only and outputs the real/fake classification and condition class classification.

our generator. The self-attention layers can extract global features from the feature map and update the configuration matrix elements according to the extracted long-range information when the neural network propagates forward, thus allowing our GAN to generate configurations with large system sizes [34, 35] (see Appendix A for an introduction to the self-attention layer).

The random matrix elements are generated from a Gaussian distribution with mean 0 and variance 1. Together with the conditional matrix, the random matrix is fed into the generator which then go through several CNN blocks and multi-head self-attention layers (Fig. 1(b)).

The neural networks map the input data into spin configurations that satisfy the temperature conditions, and generate diversified spin configurations from the random matrix. The discriminator is a classical CNN neural network, which consists of multiple CNN blocks (Fig. 1(c)). The loss functions of the generator and the discriminator are

$$
\begin{aligned}
GLoss = & E\left[-\log D_1\left(X_{\text{real}}\right) + \log\left(1 - D_1\left(X_{\text{gen}}\right)\right)\right] \\
& - E\left[Y\left(X_{\text{real}}\right)\log D_2\left(X_{\text{real}}\right)\right] \\
& - E\left[Y\left(X_{\text{gen}}\right)\log\left(D_2\left(X_{\text{gen}}\right)\right)\right],
\end{aligned}
\tag{1}
$$

and

$$
\begin{aligned}
DLoss = & E\left[-\log D_1\left(X_{\text{real}}\right) - \log\left(1 - D_1\left(X_{\text{gen}}\right)\right)\right] \\
& - E\left[Y\left(X_{\text{real}}\right)\log D_2\left(X_{\text{real}}\right)\right] \\
& - E\left[Y\left(X_{\text{gen}}\right)\log D_2\left(X_{\text{gen}}\right)\right],
\end{aligned}
\tag{2}
$$

respectively [33]. In the above equations, $D_1$ is one output branch of the discriminator which classify the input configuration into real data or generated data, and $D_2$ is another branch that determines which conditional class the input configuration belongs to, $X_{real}$ and $X_{gen}$ are the real data and the generated data, respectively, $Y$ denotes the true condition class corresponding to the $X$ fed into the discriminator, and $E[\cdots]$ represents the expectation value. It can be seen that $GLoss$ expects the discriminator to misclassify real and generated data, while $DLoss$ expects the discriminator to correctly classify the data. On the other hand, both $GLoss$ and $Dloss$ expect the discriminator to correctly classify the conditional classes of the input samples.

## 3 Generating spin configurations of the XY model

### 3.1 The 2D XY Model

The Hamiltonian of the 2D XY model is given by

$$
H = -J \sum_{\langle i,j \rangle} \cos(\theta_i - \theta_j),
\tag{3}
$$

where $J$ is the spin-spin interaction strength, $\theta_i \in (0, 2\pi]$ is the spin angle on the $i$-th site, and the sum is over all the nearest neighbours. The XY model on a square lattice is a classical model that exhibits a Kosterlitz–Thouless (KT) transition where the spin-spin correlations decays algebraically and exponentially in the low and high temperature phase, respectively. [40]. In the low-temperature phase, the vortexes and anti-vortexes stay as close to each other as possible to minimise the system's energy and tend to annihilate, causing the local winding numbers goes to zero. In the high-temperature phase, the vortexes and anti-vortexes become free. The transition takes place at $T_c = 0.89$, where the unbinding of vortex-antivortex pairs starts as temperature increases.

### 3.2 Training data and training process

We used MCMC to generate training data of the XY model with linear system size $L = 16$ and $L = 32$. To avoid critical slowing down, we sample the data in temperature region away from the phase transition point, specifically $T \in ([0, 0.8] \cup [1.4, 2])$. A total of 10,000 spin configurations are obtained and they are used as the real samples (database data in Fig.1 (a))

to train the AC-GAN.

For the discriminator, the continuous temperature conditions are divided into 10 classes with an interval equal to 0.2. Since the generator easily generates unreasonable configurations in the early stage of the training, an extra class is introduced to label the conditional class not in the temperature range $T \in ([0,2])$. In other words, we label database data with integer $1-10$ and the generated data with 11 as condition class respectively.

For each epoch, we first feed a batch of noise matrices and temperature condition (in the temperature range $T \in ([0,0.8]\cup[1.4,2])$) into the generator to produce a batch of generated configurations. Another batch of real configurations are sampled from the database and mixed with the generated configurations. Next, the discriminator is trained with the mixture data for two times. A new batch of noise matrices and temperature condition are sampled and used to train the generator twice.

The optimizer used in both the generator and the discriminator is RMSprop with a learning rate of $2.5\times10^{-4}$ and a clip-value of 0.1. The sigmoid function is used as the activation function of $D_1$ in Eq. (1) and Eq. (2), in which the output value can be any values between 0 and 1. A linear function is used as the activation function of $D_2$ and the output value ranges from 0 to 2. The kernel size of the CNN layer is specified in Fig. 1. The batch size of each epoch is 64, consisting of 32 real input data and 32 generated input data for the discriminator and 64 random input data for generator. There are totally 5000 epochs for one complete training.

## 3.3 Results

To quantify the performance of the AC-GAN, we sample spin configurations from a well-trained AC-GAN in the full temperature range and compare the physical quantities measured by the AC-GAN spin configurations with that measured by MCMC spin configurations. There are three physical quantities that are of our interests. The first one is the energy since it is the fundamental feature of a physical system. As the XY model reveals the magnetic dipole-dipole interactions between spins, the magnetization as a function of the temperature is also tested. The third physical quantity is the vorticity which quantifies the binding of vortex-antivortex pairs and gives a richer physics of the XY model as compared to the Ising model.

For a given spin configuration, the vorticity is characterised by the local winding numbers. In the continuous case, the winding number is defined as the integration over a close loop $\gamma$

$$W(\gamma) = \frac{1}{2\pi} \oint_{\gamma} (x\,dy - y\,dx), \tag{4}$$

where $x$ and $y$ represent the spin components in Cartesian coordinates. In the lattice model, we first choose a specific site $i$ and pick out the eight sites around it. The eight sites form a loop and we compute the difference in the spin orientations between the neighboring spins around this loop. The mean of the differences is then taken as the winding number of $i$, which is given by

$$W_i = \frac{1}{8\sin\frac{\pi}{4}} \sum_{j=1}^{8} \left(\sin\theta_{j+1}\cos\theta_j - \sin\theta_j\cos\theta_{j+1}\right), \tag{5}$$

where $\theta_{j+1}$ represents the spin orientation adjacent to that of the $j$-th site (i.e. $\theta_j$) in the anti-clockwise direction on the loop.

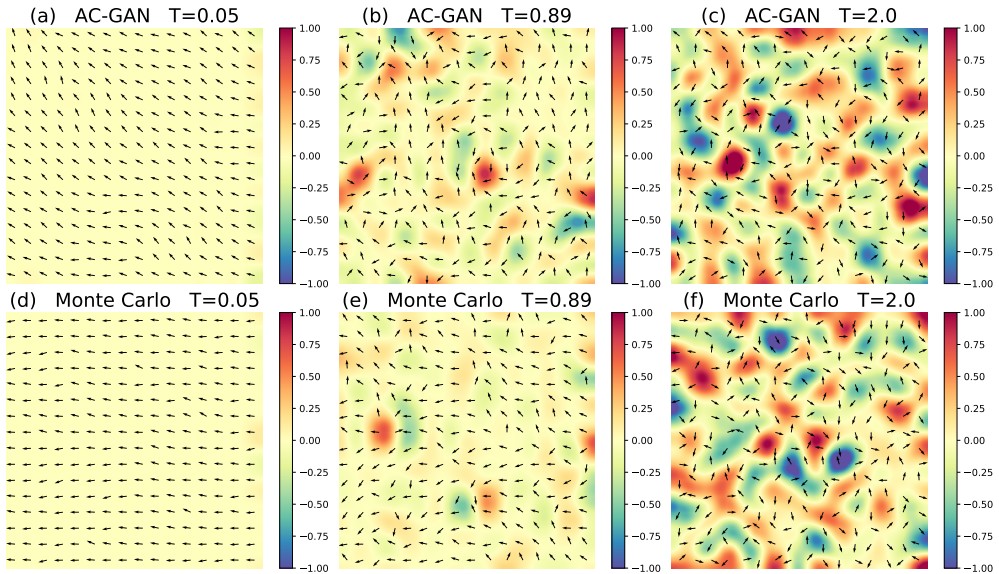

Figure 2: The spin configurations (black arrows) of XY model on a square lattice sampled from AC-GAN (left column) and Monte Carlo simulations (right column) at various temperatures. The color represents the local winding number. The spin configurations sampled from AC-GAN and Monte Carlo simulations shows consistent features. At low temperature, the spins tend to align and local winding number tends to zero anywhere. As the temperature increase across the KT transition, the spins become more disordered. The pair of the vortex (red) and the antivortex (blue) unbinds.

Figure 2 shows the spin configurations for $L = 16$ generated by GAN and MCMC respectively at low temperature (top panel), phase transition point (middle panel) and high temperature (bottom panel). The color represents the winding number discussed above. Benefiting from the self-attention layer, our GAN captures the correlation between spins and generates spin configurations with a relatively uniform orientation at low temperature, which is consistent with the spin configurations obtained from MCMC. At the phase transition point and high-temperature region, the thermal fluctuation gradually becomes severe and makes the spin configurations generated by MCMC disorder. The spin configurations generated by our AC-GAN shows similar features.

We further compare the mean energy per site, mean magnetization and the mean vorticity (absolute of winding number per site) measured by AC-GAN and MCMC, and the results are shown in Fig. 3. In the region of $T \in ([0, 0.8] \cup [1.4, 2])$, since we use a large amount of training data to train AC-GAN, the mean and variance of the observable measured by AC-GAN have an overall similar trend to those measured by MCMC. In contrast, while Implicit-GAN [32] and AC-GAN have similar performance in magnetization measurement, our AC-GAN outperforms Implicit-GAN in the other two quantities measurement, especially the vorticity (a quantitative comparison is presented in Appendix B).

We notice that in Fig. 3, the AC-GAN results are not within the error range of the MCMC results for $L = 16$ in the low-temperature region but it is consistent with MCMC results in the high-temperature region. This comes from the fact that in small systems, fluctuations from a few spins at low temperatures can significantly affect the measured macroscopic quantity. Such a situation can be improved in a larger system, and as shown in Fig. 3 (b, d, f) where

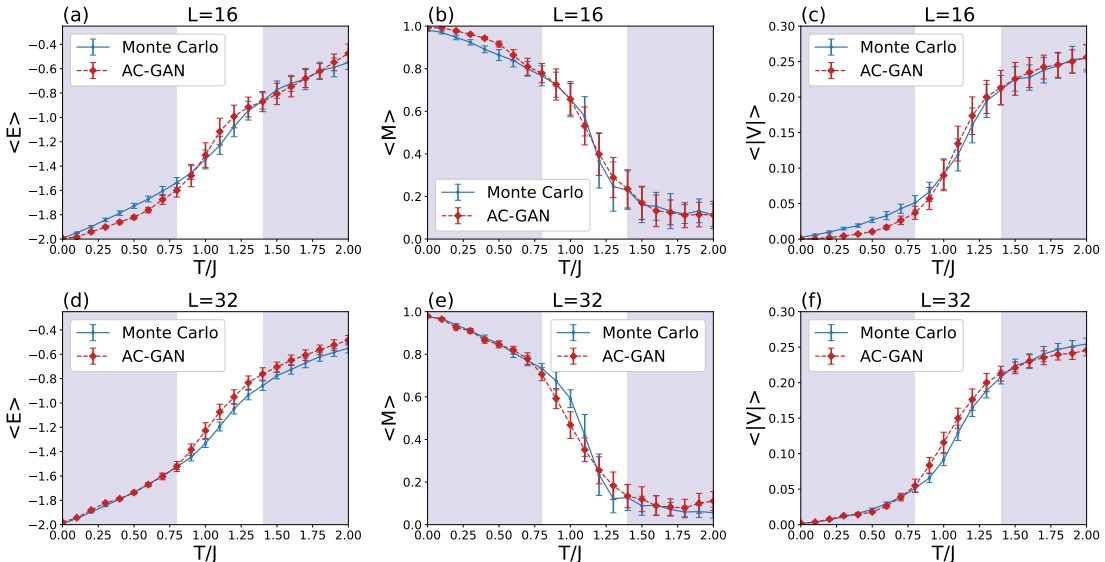

Figure 3: The mean energy density (top panel), the mean magnetization (middle panel) and the mean vorticity (bottom panel) as a function of temperature of the XY model with a linear system size of $L = 16$ (left column) and $L = 32$ (right column) measured from AC-GAN and MC. The error bars show the standard deviation of 100 data sampled by AC-GAN or Monte Carlo. The results from the two methods agree reasonably well with each other.

the system size is doubled, the AC-GAN results fall well within the error range of the MCMC results in the whole temperature range.

Moreover, it is worth noting that AC-GAN performs well at high temperatures for $L = 16$ and at low temperatures for $L = 32$. This shows that the spin configuration distribution at any temperature is learnable by the AC-GAN in principle. However, learning the spin configuration distribution of the entire temperature range in a single training can be challenging, and is in fact still an open question under active investigations in generative learning research. Using more sophisticated training techniques such as inserting spectral normalization layers between CNN layers, using multiple generators to generate data, or replacing cross-entropy loss of the discriminator by Wasserstein loss [41–43] may improve the result over the entire temperature range. Nevertheless, our results here have demonstrated the potential of using GANs to accelerate MCMC sampling.

In the vicinity of the phase transition without the training data, our AC-GAN results are also similar to that of MCMC, suggesting the deep learning model can extract information about the phase transition by learning configurations outside the critical region. Implicit-GAN obtains similar results in system sizes of $8 \times 8$ and $16 \times 16$ [32]. From Fig. 3, it is also worth noting that the performance of our AC-GAN does not deteriorate with the increase in the system size, suggesting its capability to generate configurations in larger-sized systems.

# 4 Learning with a few raw data

Using MCMC to obtain samples from large-sized 2D and most of the 3D condensed matter models requires a lot of computational power. In these cases, it is difficult to obtain a large amount of raw data through MCMC to train GANs. In this section, we present a method to obtain a large amount of training samples from just a few raw MCMC data by symmetry operations.

## 4.1 The models and the training data

We testified our scheme on the 2D XY model and 3D Heisenberg model. For the XY model, we sampled a total of 16 spin configurations with MCMC in the temperature range $T \in ([0, 0.8] \cup [1.4, 2])$ with an interval of 0.1. We then apply the following symmetry operations to each spin configuration: (1) randomly shift whole spins in xy plane, (2) reflect a spin configuration along the x-axis with 0.5 probability, (3) reflect a spin configuration along the y-axis with 0.5 probability, (4) transpose a spin configuration with 0.5 probability (random transposition), and (5) randomly rotate all spins with an angle $\theta$. A training dataset of 10,000 samples is obtained. The 2D XY model here serves as a benchmark for comparing the performance of learning with a few raw data to the results obtained from learning with a large amount of data in the previous section, allowing us to have a deeper understanding of the advantages and disadvantages of the method presented here.

We also examined whether the scheme is applicable in models at higher dimension, where obtaining a lot of configurations by MCMC becomes computationally expensive. Specially, we considered the 3D Heisenberg model [44] on a simple cubic lattice, whose Hamiltonian is given by

$$H = -J \sum_{\langle i,j \rangle} \mathbf{S}_i \cdot \mathbf{S}_j. \tag{6}$$

In the above equation, $J$ is the interaction strength between the nearest neighbouring spins and is taken to be one, $\mathbf{S}_i$ is the spin on the $i$-th site, which can be parameterised by the polar angle $\phi$ and the azimuthal angle $\theta$ as $S_i = (\sin\phi_i \cos\theta_i, \sin\phi_i \sin\theta_i, \cos\phi_i)$. The critical temperature $T_c$ for this model reported in previous work is about 1.44 [45, 46]. A total of 31 spin configurations in the temperature range $T \in ([0, 1.5] \cup [3.5, 5])$ with an interval of 0.1 are sampled from MCMC. We then apply the same symmetry operations as mentioned above for the 2D XY model to obtain 10,000 training samples.

## 4.2 Results

Figure 4 shows the results of the 2D XY model with system size $32 \times 32$ and the 3D Heisenberg model with system size $16 \times 16 \times 16$. The training process is the same as that described in Sec. 3. The observables as a function of temperature as measured from AC-GAN show a very similar qualitative behavior as that from MCMC. This evidence that a very small set of raw MCMC configurations can successfully train an AC-GAN. We expect such a data augmentation scheme can also be extended to other tasks where deep learning is used to study many-body systems to reduce the computational time for collecting training data.

Comparing the results shown in Fig. 4 (a-c) to that shown in Fig. 3 (right column), we note that the AC-GAN trained with a small dataset did not perform as well as that trained with a large number of training samples. However, we can reduce 99.84% of the time in collecting the training data using this data augmentation scheme while the results are still acceptable.

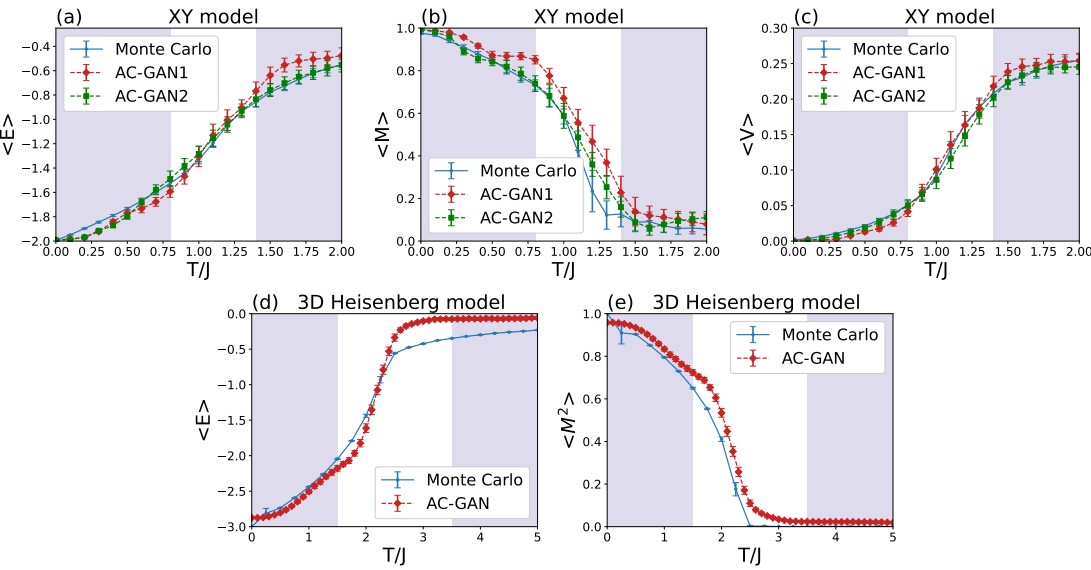

Figure 4: (a-c) show the mean energy density, mean magnetization and mean vorticity as functions of the temperature of a 2D XY model with system size $32 \times 32$ as measured by AC-GAN and MCMC. The AC-GAN1 and AC-GAN2 results were obtained using only 1 and 10 raw data respectively for each temperature interval. (d-e) show the mean energy and the mean squared magnetization as functions of the temperature of the 3D Heisenberg model with system size $16 \times 16 \times 16$. The error bars show the standard deviation of 100 data sampled by AC-GAN or Monte Carlo.

This will be particularly useful when we sample a system with a large size or strong autocorrelation where the time to collect training data will be much longer than the time to train GANs.

The main reason for the deteriorated performance is that although we obtain a large amount of training samples through data augmentation, these augmented data possess similar features which can be captured easily by the discriminator. However, it is preferred that the generator and the discriminator shall gradually improve their performance in the process of training in order to achieve the purpose of adversarial learning. The use of the augmented data greatly reduces the training difficulty of the discriminator and makes it converges to a good local minimum more rapidly than the generator, this makes the training process unstable.

Here we are showing the extreme case where only one training sample at each temperature interval are given. As we increase the number of raw data from only 1 to 10 raw data in each temperature interval, the performance of the AC-GAN increases and the sampling results from AC-GAN agree better with that from MCMC in general as shown in Fig. 4 (a-c). When applying the scheme to real research tasks, one can effectively reduce the drawback on performance by increasing the amount of raw data to strike a balance between the required computational resources and simulation time and the performance.

# 5   Detecting phase transitions by distribution difference

When a system is in a particular phase, the high-probability states following the Boltzmann distribution form a subspace in the entire Hilbert space. Upon undergoing a phase transition and transitioning to another phase, the distribution's subspace undergoes a drastic change. From the perspective of spin configurations, the correlation length experiences a significant variation. For instance, in the classical 2D Ising model at low temperatures, spontaneous symmetry breaking leads to all spins in the system aligning in one direction, resulting in a large correlation length. Conversely, at high temperatures, due to the restoration of symmetry, spins in the system randomly point up or down, leading to a small correlation length. Similarly, in the classical XY model, although true spontaneous symmetry breaking doesn't occur due to the presence of vortices, phenomena resembling spontaneous symmetry breaking emerge in finite system sizes at low temperatures. In this case, the correlation length of spins becomes substantial. When employing GANs to learn the spin configuration distribution at different temperatures, these distribution characteristics are captured. This capability allows us to use data generated by GANs to identify phase transitions effectively.

## 5.1   Definition of order parameter: distribution overlap of spin orientations

We trained our AC-GAN for 5000 epochs using the full temperature range MCMC spin configurations. The AC-GAN converges after 2000 epochs, and for the next 3000 epochs, we sampled spin configurations from AC-GAN with temperature intervals of 0.1. To measure the distribution difference, we calculated the overlap of the distribution sampled from AC-GAN and that from MCMC (to collect a sufficiently diverse dataset, we repeatedly restart the Markov chain and employ various random initial spin configurations) in each temperature interval.

Theoretically, one should calculate the overlap of the two distributions in all dimensions of the configuration space. However, for a many-body system, the dimension of the configuration space is usually high. One needs to sample a large number of configurations from AC-GAN and MCMC in order to calculate the overlap accurately but that will require a lengthy computation. Instead of calculating the overlap between the two distributions in the whole space, we randomly select two sites from the spin configurations and calculate the overlap of their distributions using the

$$\text{CrossArea}(S_i, S_j) = \frac{\min(F_{MC}(S_i, S_j), F_{GAN}(S_i, S_j))}{F_{MC}(S_i, S_j)}, \tag{7}$$

where $S_i$ is the randomly selected spin on site $i$, $F_{MC}$ and $F_{GAN}$ are the distribution of the spin configuration from MCMC and AC-GAN respectively. To minimize the error of the overlap calculated by the two sites and by the full-dimensional space, we repeated the random selections of two sites for 100 times at each temperature and calculated the average of the overlap.

## 5.2   Results

We applied our scheme of phase transition detection to the 2D square lattice Ising model with $L \in \{16, 32, 64\}$ and the 2D square lattice XY model with $L \in \{16, 32\}$ and the results are shown in Fig. 5. The data collection process and the training process of the AC-GAN are similar to that as described in Sec. 3 except that we are now using training data in the full temperature range. In Fig. 5 (a), the overlap (cross area) between the distribution of AC-GAN and MCMC is about 0.5 at low temperature, which indicates that half of the spin configurations

from MCMC are pointing up and the other half are pointing down, while all spin configurations from AC-GAN are pointing up (or all pointing down). As the temperature increases, the spin configurations of MCMC eventually become disordered and the distribution transforms into a uniform one. The overlap between the AC-GAN and MCMC distributions thus increases to a value close to one. Figure 5 (c) shows the first derivative of the CrossArea with respect to the temperature. The derivative shows a significant change around the transition. From the maximum of the gradient, we estimated the transition temperature to be 2.5, 2.4 and 2.3 for $L = 16, 32$, and 64, respectively. The estimated transition temperature tends to the theoretical value as the system size increases.

Figure 5 (b) shows the results of the 2D XY model. When the system is at low temperature, the configurations sampled by MCMC can have spins lining up in any direction. As mentioned above, this linear distribution of two spins in multidimensional space can cause AC-GAN to experience a symmetry-breaking-like phenomenon. As a result, the configurations sampled by AC-GAN will have spins line up in one direction while the input noise only cause small local fluctuation in the spins. Therefore, the overlap between MCMC data and AC-GAN data is low. When the temperature of the system gradually increases across the phase transition, the distribution of MCMC data gradually tends to a uniform distribution, which is a friendly and easy-to-fit distribution for AC-GAN. Thus, the AC-GAN will not experience the symmetry-breaking-like phenomenon, and the resulting overlap between the MCMC and the AC-GAN data is high. The overlap converges to about 0.5 above $T = 1.2$. It is worth noting that the overlap tends to 0.5 at high temperatures does not mean that AC-GAN experiences the symmetry-breaking-like phenomenon. Instead it is caused by an insufficient number of samples when we calculate the overlap of two distributions. While the spins of the Ising model are binary and we can accurately calculate the overlap, the spins of the XY model can have orientation ranged $(0, 2\pi]$. This requires more samples to better reflects the distribution of MCMC data and AC-GAN data on the 2-dimensional plane. If the number of samples increases, the overlap is expected to converge to a value closer to one. Nevertheless, to detect the phase transition, we only care about if there is a significant change in the overlap as a function of the driving parameter but not the value to which the overlap converges. Figure 5 (d) shows the first derivative of the CrossArea with respect to the temperature and the estimated phase transition point from the maximum gradient is $T_c = 1.0$ and $T_c = 0.9$ for $L = 16$ and $L = 32$, respectively. Note that similar to other unsupervised machine learning methods of phase transition detection [18, 19, 21, 47–51], a detailed error estimation on the predicted $T_c$ is not available. However, the convergence of the predicted $T_c$ as the system size increases suggests the validity of the method.

Finally, we would like to remark that the method proposed here does not require any preprocessing of the data and it is a universal method for locating phase transitions. It is also different from other machine learning methods in detecting phase transitions where the machine works as a black-box. Our method has a physical interpretation that the phase transition is accompanied by the change in the configuration distribution.

## 6  Conclusion

In this work, we investigated the application of GANs in many-body systems. First, we found that AC-GAN with self-attention layers can better capture the spin-spin correlation in many-body systems and generate high-quality configurations. We tested our deep learning model on the 2D XY model. The spin configurations sampled by AC-GAN were almost identical to that

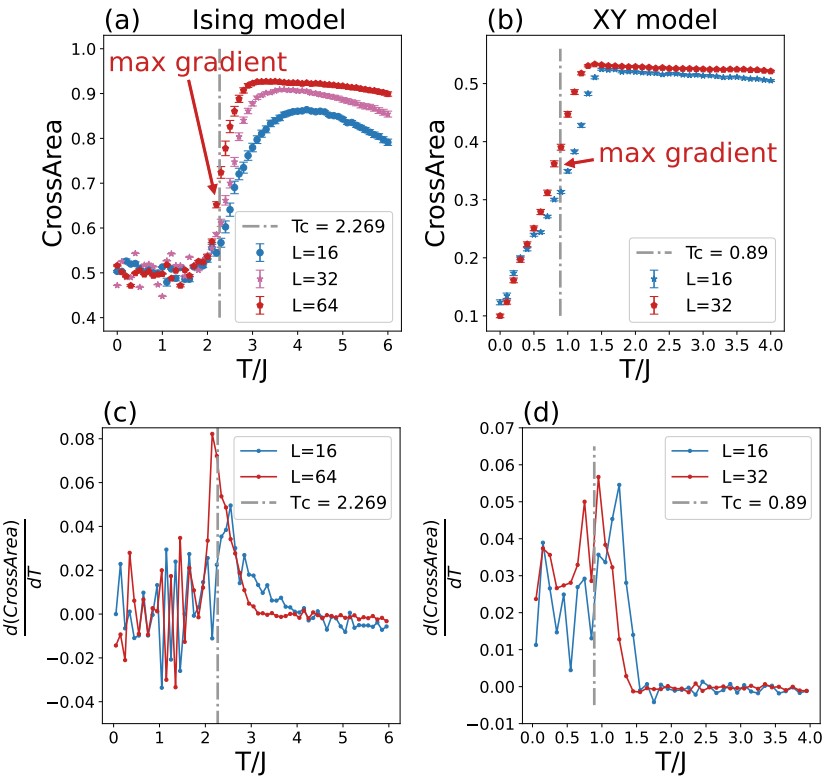

Figure 5: Cross-area as a function of temperature in (a) the 2D Ising model and (b) the 2D XY model with various system sizes. (c) and (d) is the first derivative of the Cross-Area in (a) and (b) with respect to temperature, respectively. The transition temperature estimated by the maximum of the gradients for the largest system agrees well with the theoretical transition temperature (vertical dash-dotted line). The error bars represent standard deviations of 100 samples.

sampled by MCMC. The calculated mean energy, magnetization and vorticity from AC-GAN also agree well with that calculated from MCMC.

We further examined the performance of AC-GAN trained with only a few raw data. We used MCMC to sample a few spin configurations and apply data augmentation on these configurations using symmetry operations. The AC-GAN can be successfully trained with the augmented data and it performed well. The success of this data augmentation method not only allows us to study condensed matter models which have difficulties in MCMC sampling through AC-GAN but also suggests that the method can be applied to other deep learning tasks where a large number of configurations is required.

We also found that the the symmetry-breaking-like phenomenon in AC-GAN can be used to locate the phase transitions. We first used full temperature range MCMC data to train AC-GAN. By calculating the overlap between the distribution of configurations sampled by AC-GAN and MCMC, we measured the distribution difference and further locate the phase transition point of the system. The critical points in the 2D Ising model and the XY model on a square lattice are successfully determined. Such an approach of phase transition detection provide us physical insights into the information contained in the configuration space itself instead of the order parameters. One may apply the scheme to more complicated systems such as the random bond model [52] in which the order parameter is still unknown or it is hard to calculate the order parameter accurately using Monte Carlo simulations to study phase transition.

# Acknowledgements

We acknowledge financial support from National Natural Science Foundation of China (Grant No. 12005179, 12204130), Research Grants Council of Hong Kong (Grant No. CityU 11318722), City University of Hong Kong (Grant No. 9610438, 7005610, 9680320), and Harbin Institute of Technology Shenzhen (Grant No. ZX20210478, X20220001).

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

## A   Self-attention layers

In the classical convolutional GAN, the decoder and encoder in the generator are multi-layer convolutional neural network (CNN). The performance of this GAN model will be limited by the kernel size of the CNN layers. Even though the receptive field of the kernel increases as the encoder deepens, the kernel only operates on the matrix elements within its size in the feature map and information from elements far away is ignored. However, long-range correlations can play a significant role in condensed matter models. If the generator is unable to capture this long-range information in the feature map, the generated spin configurations will be of low quality. To overcome this deficiency, we introduced the self-attention layers into the network. The self-attention layers can learn the global relationship between the elements and further adjust the value of each element in the feature map, thus allowing the generator to fine-tune the spin on each site in the generated configurations according to the long-range correlation.

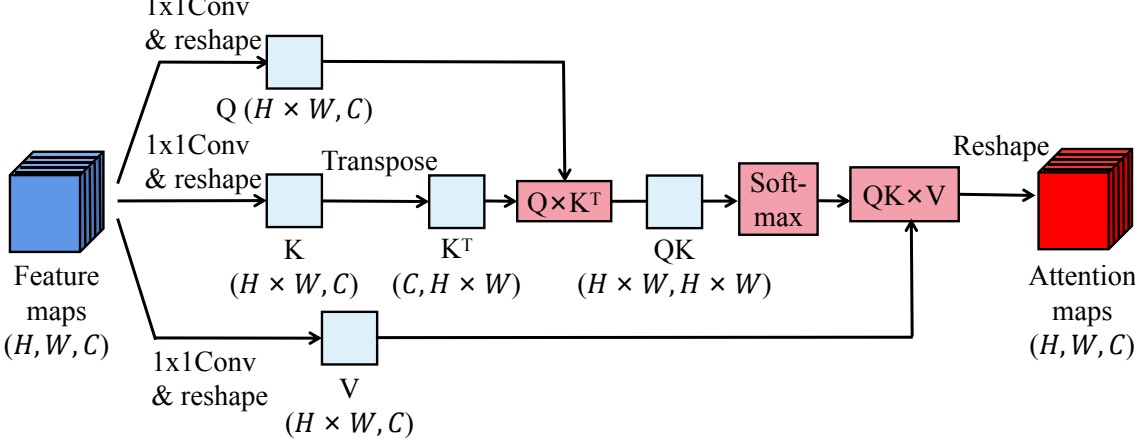

Figure 6: The architecture of the self-attention layer.

The architecture of the self-attention layer is shown in Fig.6. Specifically, the feature maps input to the self-attention layer is a 3D tensor with a shape of height, width, and channel $(H, W, C)$. The feature maps are fed into three CNN layers with a kernel size of 1x1 and reshaped into $(H \times W, C)$, and we get three matrices named query (Q), key (K) and value (V) respectively. The transpose of K is multiplied by Q to obtain a matrix QK, which is also called the energy matrix, with shape $(H \times W, H \times W)$. The $(i, j)$-th element in the energy matrix represents the relationship between the $i$-th element in the feature map on all channels and the $j$-th element on all channels (Note that here we only use one index to define the position of an element in the feature map since we have flatten each feature map into a 1D vector). By applying the softmax activation function on QK, all elements of each row of QK now add up to one. This re-scaling makes the self-attention layer converge faster. Finally, QK is multiplied by V to incorporate the channel information. The resultant matrices are then reshaped into a 3D tensor having the same shape as the input feature maps.

# B  Comparing the performance between AC-GAN and Implicit-GAN

To compare the performance of AC-GAN and Implicit-GAN quantitatively, we computed the percentage difference between the GAN-generated results and the MCMC results in the XY model. Figure 7 shows the results for measurements on various physical quantities. While both GANs perform similarly in magnetization measurements (Fig. 7(b)), the AC-GAN surpasses the Implicit-GAN in the energy and vorticity measurements. Specifically, in the energy measurement, the Implicit-GAN gives a percentage difference that is about two times of our AC-GAN in the high-temperature regime (Fig. 7(a)). Besides, in the calculation of vorticity, the Implicit-GAN performs very poorly in the low-temperature regime, where the percentage difference is generally higher, and can be 10 times higher, as compared to the case of using our AC-GAN (Fig. 7(c)). This provides strong evidence that our AC-GAN performs better than Implicit-GAN.

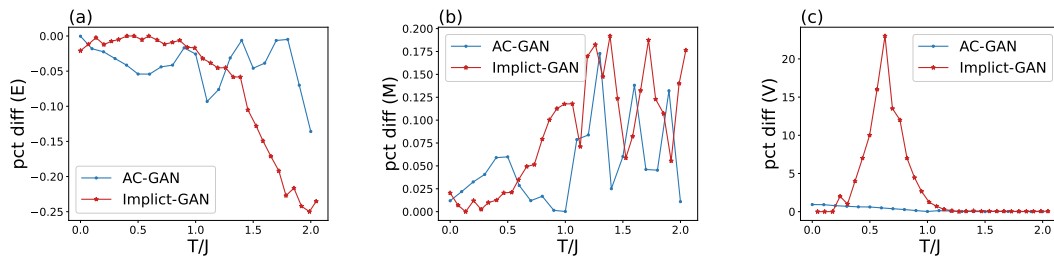

Figure 7: The percentage difference in (a) the energy, (b) the magnetization, (c) the vorticity between the GAN and MCMC results for the XY model with system size $L = 16$. Our AC-GAN generally gives significantly smaller percentage difference in the energy measurement at high temperatures and vorticity measurement at low temperatures.