# Peer review of "Rapid Measurements and Phase Transition Detections Made Simple by AC-GANs"

_SciPost Physics Core_

## Round 1 · Referee Report · Anonymous (Referee 1) · 2024-3-3

Report
Unfortunatly, non of the stated expectations of SciPost physics is matched:
Expectations (at least one required) - the paper must:
Detail a groundbreaking theoretical/experimental/computational discovery;
Present a breakthrough on a previously-identified and long-standing research stumbling block;
Open a new pathway in an existing or a new research direction, with clear potential for multipronged follow-up work;
Provide a novel and synergetic link between different research areas.
Thus, although results seem consistent and the research done has value, I believe this article is not suitable to publish in SciPost physics.
Many of the concepts and or techniques used in the present article are equal or small modifications of https://www.scipost.org/SciPostPhys.11.2.043/pdf Moreover, the clarity of the written English is far from acceptable, and more details of the network architectures should be added.

---

## Round 1 · Referee Report · Anonymous (Referee 2) · 2024-3-25

Strengths
2-addresses important question
3-valuable results
Weaknesses
Report
In the manuscript with title “Rapid Measurements and Phase Transition Detection Made Simple by AC-GANs”, Ding, Tang, and Yu study how conditional generative adversarial networks (GANs) with auxiliary classifiers can be used to sample from classical spin models. The models studied are the two-dimensional XY model as well as the Heisenberg model in three spatial dimensions. They find that their method outperforms related previous GAN-based approaches. They further discuss how performing symmetry transformations might be used to obtain better results for small number of samples and also present a way of detecting phase transitions.
I think the manuscript is very well written and presented. It also addresses an important question – how to improve sampling algorithms using generative models – and, as far as I can see, seems to add a valuable contribution. As such, I recommend publication of the work in SciPost physics, in principle. However, I first would like to ask the authors to address the following points:
1) The authors only use simple observables to evaluate the performance of the models. If possible, it could be worthwhile to include other measures such as earth mover distances of the distributions.
2) Did the authors test whether their approach suffers from mode collapse – a well-known issue of GANs?
3) I do not fully understand in which sense using Eq. (7) is beneficial compared to other ways people have studied for the detection of phase transitions. It seems, one needs both MCMC data and train a GAN. Relatedly, in the last paragraph before the conclusion starts, the authors write “It is also different from other machine learning methods in detecting phase transitions where the machine works as a black-box. Our method has a physical interpretation that the phase transition is accompanied by the change in the configuration distribution.” I likely do not understand their approach well enough, but I am not so sure in which sense their approach is less of a black-box than other machine-learning techniques studied before and I think previous approaches are also based on changes in the configuration distributions, see, for instance, Phys. Rev. E 99, 062107 (2019) or also that in Ref. 32.
4) I also noticed a few typos: (i) page 2, “To let the Generator performs …”, (ii) page 3, “… besides the random matrix z, constrains are added …”, and (iii) in the caption of Fig. 2, it seems that column and rows are swapped.

---

## Round 2 · Referee Report · Anonymous (Referee 2) · 2024-5-4

Report

I have read the response of the authors to my questions and studied the revised manuscript. Concerning C3/R3: I still believe that the author's approach is as much of a "black box" as at least some other machine learning detection techniques for phase transitions, including those mentioned. However, since the authors removed the statement about the "black box" from the manuscript, I think their response/changes is satisfactory. Also the other parts of the response and the changes are fine in my opinion.

As such, I think the work can be published in SciPost in its current form.

Recommendation

Publish (meets expectations and criteria for this Journal)

---

## Round 2 · Referee Report · Anonymous (Referee 1) · 2024-5-5

Strengths

1- The authors improved performance with respect to previous work for the same task, specifically in the quality of the data generation, through the addition of self attention layers mainly.

2- They proposed a different way to see phase transitions using both the network and the monte carlo data which is intuitive, although noisy.

Weaknesses

1- This work follows *exactly*** the same structure and scope of https://scipost.org/SciPostPhys.11.2.043

There, the authors measure the same observables on the same system, also discuss enhance sampling applications, also play with symmetries (although in a different way), and also have a network-way to spot phase transitions, using what they defined as GAN fidelity.

2- The clarity of the presentation is not great, nor that of written english. There are simply too many empty phrases trying to make the results appear novel and outstanding. Moreover there are phrases like "The neural networks map the input data into spin configurations that satisfy the temperature conditions" which are simply meaningless.

Report

Dear authors,

concerning your comment " We sincerely suggest the referee read our manuscript in more detail."

I must say that I carefully read your paper once, and now I did it yet again, reaching precisely the same conclusion.

Your work is a consistent, correct and natural way to improve or extend the previous work on GANs https://scipost.org/SciPostPhys.11.2.043.

This, nonetheless, does not match the scope of scipost, that I already mentioned in my previous report.

Reply to your replies:
R1: "First, we propose the advanced integration of self-attention layers, which significantly improves the performance of the GAN model. Especially some of the physical quantities such as the vorticity, can be estimated with much higher accuracy, as shown in Fig. 8c in Appendix A of our manuscript. "

RR1: This only reinforces my claim about your paper being a modification of the cited one. Adding a self attention layer to a network is not a break-through, is a standard procedure in any application of contemporary computer science.

"[...]Second, we also discuss how the symmetries of the lattice can be utilized to augment the training dataset, allowing us to train GANs more efficiently with much less data."
RR2: Symmetries were also discussed in the cited article, and applying transformations to do data augmentation is yet again nothing but a standard thing to do.

"[...] Third, we proposed a novel quantity of distribution overlap to capture the phase transition. The quantity provides a physically intuitive way to detect the phase transition point, and it is especially useful when the order parameter in the model is unknown."

RR3: The cited article also provides one (which is actually independent of the monte carlo data, which could be considered an advantage), you proposed another, which although interesting proved to be quite noisy.

I'm sorry, I can only say that you did a good work, which could be more suitable for e.g. PRE or PRB.

Best regards.

Recommendation

Reject

---

## Round 2 · Author Response

Response to the second referee:
We thank the referee for the careful reading of our manuscript and his/her support on publication of our work. Below are our responses (R) to the referee's comments (C).

C1: The authors only use simple observables to evaluate the performance of the models. If possible, it could be worthwhile to include other measures such as earth mover distances of the distributions.
R1: We thank the referee for the suggestion. In the revised manuscript, we have added the discussion of the Earth mover's distance and the Jensons-Shannon divergence in Appendix C. Figure 9 shows the Earth Mover’s distance and Jenson-Shannon (JS) divergence measuring the difference in the distribution of the MCMC samples and the AC-GAN generated samples. Both the distance measures are small throughout the simulated temperature range, providing further evidence of the validity of our method.

C2: Did the authors test whether their approach suffers from mode collapse – a well-known issue of GANs?
R2: Our model also suffers from certain degree of mode collapse depending on the system’s driving parameters. Nevertheless, as shown in Fig.9 of the revised manuscript, the small earth mover distance and JS divergence suggest the sampled distribution from GANs is sufficiently close to that from MCMC. Therefore, the mode collapse did not affect the use of GAN to sample physical quantities. We have added a comment on this point in the last paragraph of Sec. 3 of the revised manuscript.

C3: I do not fully understand in which sense using Eq. (7) is beneficial compared to other ways people have studied for the detection of phase transitions. It seems, one needs both MCMC data and train a GAN. Relatedly, in the last paragraph before the conclusion starts, the authors write “It is also different from other machine learning methods in detecting phase transitions where the machine works as a black-box. Our method has a physical interpretation that the phase transition is accompanied by the change in the configuration distribution.” I likely do not understand their approach well enough, but I am not so sure in which sense their approach is less of a black-box than other machine-learning techniques studied before and I think previous approaches are also based on changes in the configuration distributions, see, for instance, Phys. Rev. E 99, 062107 (2019) or also that in Ref. 32.
R3: We thank the referee for the questions. The order parameters that detect the phase transitions in condensed matter models may not be always known and can be difficult to obtain. Therefore, in most cases, even with extensive Monte Carlo data, it is challenging to locate the transition points accurately. By using Monte Carlo data, AC-GANs data, and Equation 7, our method can effectively locate the transition point without the need of the order parameters, thus demonstrating the practical value of our approach. We have added a comment on this point in the last paragraph of Sec.5.1 in the revised manuscript.

The term ‘black box’ generally refers to the issue that deep learning methods have, i.e., what makes the machine effectively learn the phase transition is unknown. In Physical Review E 99, 062107 (2019), the authors use confusion of the machine for phase transition detection and provide a rigorous and mathematical definition of the confusion. However, the reason for why the machine experiences confusion near the transition point was not addressed. In comparison, our method does not rely on confusion but on the changes in the distributions during the learning process to detect the transition. It provides a more physically intuitive way to capture the phase transition from the perspective of configuration distributions. Nevertheless, to avoid confusion, we have removed the statement about the ‘black box’, and further explain the benefits of our approach in the last paragraph of Sec. 5.2.

C4: I also noticed a few typos: (i) page 2, “To let the Generator performs …”, (ii) page 3, “… besides the random matrix z, constrains are added …”, and (iii) in the caption of Fig. 2, it seems that column and rows are swapped.
R4: We thank the referee for carefully reading our manuscript and pointing out these typos. We have corrected them in the revised manuscript.

Response to the first referee:
Below are our responses (R) to the referee's comments (C).
C1: although results seem consistent and the research done has value, I believe this article is not suitable to publish in SciPost physics.
Many of the concepts and or techniques used in the present article are equal or small modifications of https://www.scipost.org/SciPostPhys.11.2.043/pdf
Moreover, the clarity of the written English is far from acceptable,
R1: We strongly disagree with the referee’s comment that our work is just a minor modification of SciPost Phys. 11, 043 (2021) and his criticism on the presentation of our manuscript. The second referee also commented that our manuscript is “very well written and presented. It also addresses an important question – how to improve sampling algorithms using generative models.”. Our work is distinguished from the mentioned work in the following ways. First, we propose the advanced integration of self-attention layers, which significantly improves the performance of the GAN model. Especially some of the physical quantities such as the vorticity, can be estimated with much higher accuracy, as shown in Fig. 8c in Appendix A of our manuscript. Second, we also discuss how the symmetries of the lattice can be utilized to augment the training dataset, allowing us to train GANs more efficiently with much less data. Third, we proposed a novel quantity of distribution overlap to capture the phase transition. The quantity provides a physically intuitive way to detect the phase transition point, and it is especially useful when the order parameter in the model is unknown. We sincerely suggest the referee read our manuscript in more detail.

C2: more details of the network architectures should be added.
R2: In the revised manuscript, we have included a more detailed architect of the deep learning model as Fig.7 in Appendix A to further clarify our methodology.

---

## Round 2 · List of Changes

1. Added Appendix C, which discusses the difference in the distribution of energy sampling between MC data and AC-GAN data as measured by Earth Mover’s Distance and Jensons-Shannon Divergence.
  2. Added a remark in the last paragraph of Sec.3 to discuss that mode collapse does not affect the effective sampling of physical quantities by the deep learning model.
  3. Added a comment in the last paragraph of Sec. 5.1 regarding the practical value of Equation 7 for locating phase transitions.
  4. Modified the mentions of black-box in the introduction and conclusion to avoid confusion, and further explain the benefits of our approach in the last paragraph of Sec. 5.2.
  5. Added the detailed architecture of the deep learning model used in this work as Fig. 7 in Appendix A.
  6. Corrected the typos.

---

## Round 3 · List of Changes

1. Typeset the captions in Figure 4 and Figure 5.
2. Added DOI to the references.
3. Clarified some statements to improve the presentation.
4. Corrected typos and grammatical errors.

---

## Editorial Decision

editorial_decision: